# Pollution Characteristics and Human Health Risk Assessment of Heavy Metals in Street Dust from a Typical Industrial Zone in Wuhan City, Central China

**DOI:** 10.3390/ijerph191710970

**Published:** 2022-09-02

**Authors:** Hong Chen, Changlin Zhan, Shan Liu, Jiaquan Zhang, Hongxia Liu, Ziguo Liu, Ting Liu, Xianli Liu, Wensheng Xiao

**Affiliations:** 1School of Environmental Science and Engineering, Hubei Polytechnic University, Huangshi 435003, China; 2Hubei Key Laboratory of Mine Environmental Pollution Control and Remediation, Hubei Polytechnic University, Huangshi 435003, China; 3College of Environment, Zhejiang University of Technology, Hangzhou 310014, China; 4State Key Laboratory of Loess and Quaternary Geology, Institute of Earth Environment, Chinese Academy of Sciences, Xi’an 710061, China

**Keywords:** heavy metal, street dust, pollution, critical source, health risk assessment

## Abstract

This study aimed to assess the pollution levels, sources, and human health risks of heavy metals in street dust from a typical industrial district in Wuhan City, Central China. In total, 47 street dust samples were collected from the major traffic arteries and streets around Wuhan Iron and Steel (Group) Company (WISC) in Qingshan District, Wuhan. The concentrations of heavy metals (Cr, Mn, Ni, Zn, Fe, Cu, and Cd) in street dust were determined by atomic absorption spectroscopy. Results indicated that the mean concentrations of Zn (249.71 mg/kg), Cu (51.15 mg/kg), and Cd (0.86 mg/kg) in street dust were higher than their corresponding soil background values in Hubei Province. Heavy metal enrichment is closely related to urban transportation and industrial production. The pollution level of heavy metals in street dust was assessed using the geo-accumulation method (*I*_geo_) and potential ecological risk assessment (PERI). Based on the *I*_geo_ value, Cr, Mn, Fe, and Ni showed no pollution, Zn and Cu showed light to moderate contamination, and Cd showed moderate contamination. The PERI values of heavy metals in street dust ranged between 76.70 and 7027.28, which represents a medium to high potential ecological risk. Principal component analysis showed that the sources of heavy metals in street dust were mainly influenced by anthropogenic activities. Among the studied metals, Cu, Cr, Zn, Fe, and Mn mainly come from industrial processes, while Ni and Cd come from traffic exhaust. The non-carcinogenic risk indexes of heavy metals for children and adults are ranked as Cr > Cu > Ni > Cd > Zn. The health risks to children through the different exposure pathways are higher than those for adults. Hand-to-mouth intake is the riskiest exposure pathway for non-carcinogenic risk. In addition, Cr, Ni, and Cd do not pose a carcinogenic risk for the residents.

## 1. Introduction

With the rapid development of industrialization and urbanization, the city has become the most active area for human activities. However, the urban environment is largely affected by various factors, such as industrial production, transportation, construction, and residents’ daily life. As a typical environmental medium, street dust is an important reflection of the concentration and spatial distribution of heavy metal pollutants in the immediate environment [1,2]. Meanwhile, due to the high surface area of street dust, it is also an important carrier for organic and inorganic pollutants, especially heavy metals [3]. Street dust acts as the source and sink of heavy metals and also receives heavy metals from atmospheric deposition. Due to the toxicity, cumulative trends, and non-degradable potential of heavy metals in street dust [4], it has attracted widespread attention. The pollution sources of heavy metals are generally divided into two types, i.e., natural and anthropogenic sources. Human activities have been proved to be the major sources of heavy metal pollution, such as metallurgic processing, traffic emissions, chemical fertilizers, waste disposal, the construction of buildings and roads, etc. [5,6,7,8,9].

Previous studies have shown that urban street dust is polluted to varying degrees by heavy metals [10,11,12,13,14,15,16,17,18]. Moreover, street dust can come into direct contact with the human body and produce hazards to human health [19,20], increasing the risk of Alzheimer’s disease [21,22], cardiovascular disease [23,24], and atherosclerosis [25,26]. In addition, the composition and concentrations of heavy metals in urban dust can directly reflect the long-term or short-term human activities in the region. Heavy metals in dust particles, especially fine particles, are re-suspended into the atmosphere, which can cause health risks for local residents via ingestion, inhalation, or dermal exposure [27]. Previous studies found that among the three exposure pathways, hand-to-mouth ingestion is the most important exposure pathway for human beings to absorb street dust, thus causing certain hazards to the human body, involving both non-carcinogenic risk and carcinogenic risk [27,28]. 

Air pollution in China’s cities has become a serious problem in the last ten years, which is closely related to the city’s economic stability, human health and safety, and social stability. Secondary suspension of street dust has become an important source of air pollution [29,30]. In addition, the absorption of heavy metals in organisms largely depends on their chemical speciation. At present, there are many studies about heavy metal pollution in street dust that have been conducted in China, involving pollution characteristics, occurrence forms, environmental magnetic response, source analysis, particle size effect, and health risks [13,14,15,16,17,18,31,32,33]. 

Qingshan District (QSD) is known as an important steel production area with a large chemical industry and is also a heavy industry zone in Wuhan City, Central China. After more than 50 years of construction and development, it has national significance for industries such as steel manufacturing, petrochemicals, and equipment manufacturing. However, with the rapid development of urbanization and industrialization, the environmental pollution problem in QSD has become more and more serious. The effects of urban street dust on the local environment and human health have increased continuously, but there is less relevant research available that is specific to QSD, Wuhan City. Therefore, the objectives of this study were: (1) to investigate the pollution levels of Cr, Mn, Ni, Zn, Fe, Cu, and Cd in street dust, and reveal their spatial characteristics; (2) to evaluate the potential ecological risk and human health risk of heavy metals in street dusts; (3) to analyze the potential sources of heavy metals in street dust using principal component analysis. The study results provide a scientific basis for the environmental protection of local residents and the construction of an eco-city.

## 2. Materials and Methods

### 2.1. Study Area

Qingshan District (30°37′ N, 114°26′ E), one of the central urban areas of Wuhan, is also an important industrial zone, with a population of over 0.45 million in Wuhan City. The district is located in northeastern Wuhan City, adjacent to Wuchang District in the west and Hongshan District in the south, surrounded by the Yangtze River to the east and north. It has become an important industrial town, mainly due to its eight pillar industries: metallurgy, chemical manufacture, environmental protection, electric power, machinery, shipping, construction, and building materials. QSD is located in the transportation loop between the central and outer rings of Wuhan, connecting the Beijing, Zhuhai, Shanghai, and Chengdu high-speed trunk lines, and includes the Wudong railway marshaling station and multiple freight terminals on the Yangtze River Golden Waterway. There are 110 roads in QSD with a total length of 137.15 km. The per capita road area is 6.5 square meters, and the road network density in the area is 3.05 km per square kilometer. Among them, the four main roads represent a transportation hub connecting the three towns of Wuhan and leading to all parts of the country. 

### 2.2. Sample Collection and Analysis Method

In this study, a total of 47 street dust samples were collected in QSD during May 2018 (Figure 1). At each sampling site, a polyethylene brush and dustpan were used to sweep both sides of the road, repeated at least 3 times to collect about 100 g of dust. The dust samples were stored in numbered self-sealing polyethylene bags and then transported to the laboratory. All the samples were air-dried for 5 days and then sieved through a 1.0 mm nylon mesh to remove debris. After the samples were carefully homogenized and passed through a 100 μm sieve, the sieved samples were stored at 4 °C before analysis. 

About 0.2 g of dust samples were weighed in a Teflon crucible and then digested with HNO_3_, HF, HClO_4_, and HCl at a temperature of 180–350 °C using an electro-thermal plate. Each sample solution was filtered, using a 0.45-μm membrane filter, into a 50 mL volumetric flask and diluted with HNO_3_ (2%, *v/v*). The concentrations of heavy metals (Cr, Mn, Ni, Zn, Fe, Cu, and Cd) were analyzed using atomic absorption spectrometry (FAAS, Varian AA240, NJ, USA). A blank control was added to each group of digested samples, and a soil standard reference material sample (GSS-5, National Research Center for Certified Reference Materials, Beijing, China) was added to every 10 samples. The recovery rate of heavy metals was within the range of 84.5–116.4%. The relative standard deviations of replicated samples were lower than 10%.

### 2.3. Pollution Assessment Methods

#### 2.3.1. Geo-Accumulation Index (*I*_geo_)

The geo-accumulation index (*I*_geo_) is widely used to assess the contamination of street dust by comparing the levels of heavy metals obtained in samples to the background levels [27,29]. The calculation formula is shown in Equation (1):(1)$$I_{\mathrm{geo}}=\log_{2}\left(\frac{C_{n}}{1.5B_{n}}\right)$$
where *C*_n_ (mg/kg) is the concentration of each metal in street dust and *B*_n_ (mg/kg) is the geochemical background concentration of each metal. The constant, 1.5, is a correction index that is commonly used to offset the effect of natural variation [29]. According to the author of [34], the values are: uncontaminated (*I*_geo_ ≤ 0); uncontaminated to moderately contaminated (0 < *I*_geo_ ≤ 1); moderately contaminated (1 < *I*_geo_ ≤ 2); moderately to heavily contaminated (2 < *I*_geo_ ≤ 3); heavily contaminated (3 < *I*_geo_ ≤ 4); heavily to extremely contaminated (4 < *I*_geo_ ≤ 5); and extremely contaminated (5 < *I*_geo_). 

#### 2.3.2. Potential Ecological Risk Assessment (*PER**I*)

The potential ecological risk index (*PER**I*) was established for evaluating heavy metal pollution and ecological hazards based on the principles of sedimentation [35]. The method considers various factors, such as multi-element synergy, toxicity level, pollution concentration, and environmental sensitivity to heavy metals, and is widely used in the environmental risk assessment of heavy metals [11,19]. The calculation formulas are shown in Equations (2)–(4):(2)$$C_{f}^{i}=\frac{C_{r}^{i}}{C_{n}^{i}}$$
(3)$$E_{r}^{i}=T_{r}^{i}\times C_{f}^{i}$$
(4)$$PERI=\sum_{i=1}^{n}E_{r}^{i}$$
where *C_f_^i^* is the calculated contamination factor; *C_r_^i^* (mg/kg) is the measured concentration of each metal, and *C_n_^i^* (mg/kg) is the geological background value of each metal; *E_r_^i^* is the potential ecological risk of each toxic metal; *T_r_^i^* is the toxicity coefficient of heavy metals (Cr = 2, Ni = 5, Mn = 10, Cd = 30, Cu = 5, Zn = 1) [35]; *PERI* is the combined potential ecological risk index for multiple metals. The classification of pollution level and ecological risks according to Hakanson’s method are shown in Appendix A in the Appendix A.

### 2.4. Human Health Risk Assessment

Children and adults are exposed to three major forms of health risks with dust-based heavy metals, namely, hand-to-mouth ingestion, respiratory ingestion, and skin exposure. In this study, the health risk assessment model recommended by the US Environmental Protection Agency [36] was used to assess the health risk of five heavy metals (Cr, Ni, Cd, Cu, and Zn) with chronic non-carcinogenic risk. The long-term daily average exposures of the three exposure pathways and the exposure to carcinogenic heavy metals can be calculated as follows [37,38]:(5)$$ADD_{ing}=C\times \frac{IngR\times EF\times ED}{BW\times AT}\times 10^{-6}$$
(6)$$ADD_{inh}=C\times \frac{InhR\times EF\times ED}{PEF\times BW\times AT}$$
(7)$$ADD_{dermal}=C\times \frac{SA\times AF\times ABS\times EF\times ED}{BW\times AT}\times 10^{-6}$$
(8)$$LADD_{inh}=\frac{C\times EF}{PEF\times AT}\left(\frac{InhR_{child}\times ED_{child}}{BW_{child}}+\frac{InhR_{adult}\times ED_{adult}}{BW_{adult}}\right)$$
where *ADD_ing_* [mg·(kg·d)^−1^], *ADD_inh_* [mg·(kg·d)^−1^], and *ADD_dermal_* [mg·(kg·d)^−1^] are oral intake, respiratory inhalation, and skin contact pathways, respectively; *C* (mg/kg) is the concentration of heavy metals in street dust; *IngR* (mg/day) represents the rate of street dust ingestion; *EF* (day/year) is the exposure frequency; *ED* (year) is the exposure duration; *InhR* (m^3^/day) represents the rate of human inhalation of air containing street dust; *SA* (cm^2^) is the surface area of human skin which is exposed to dust; *AF* and *ABS* are the dermal adherence and absorption factors, respectively; *PET* (m^3^/kg) is the particle emission factor; *BW* (kg) is the average body weight of the exposed individuals; *AT* is the average time of exposure with the pollutants, in days. The values of the above parameters were adopted from [39,40] and are listed in Appendix A in the Appendix A.

Non-carcinogenic risk (*HI*) and carcinogenic risks (*CR*) were calculated using Equations (9)–(11) [41]:*HQ* = *ADD*/*RfD*(9)
(10)$$HI=\sum HQ_{i}\;$$
*CR* = *LADD* × *SF*(11)
where *ADD* [mg·(kg·d)^−1^] is the daily average exposure dose; *RfD* [mg·(kg·d)^−1^] represents the daily reference ingested dose of the contaminant via an exposure route; *HI* is the hazard index; *CR* represents the probability of cancer occurrence; and *SF* [mg·(kg·d)^−1^] is the carcinogenic slope factor. The *RfD* and *SF* values of the different metals participating in the health risk assessment are shown in [40].

If the *HQ* or *HI* value < 1, the risk is considered relatively small or negligible; if the value of *HQ* or *HI* > 1, the non-carcinogenic risk is considered to be significant. It is generally believed that if *CR* < 10^−6^, the substance is not considered to be carcinogenic; if *CR* ranged from 10^−6^ to 10^−4^, an acceptable carcinogenic risk is assumed; if *CR* > 10^−4^, an unacceptable risk of carcinogenicity is assumed [40]. 

## 3. Results and Discussion

### 3.1. Heavy Metal Pollution in Street Dusts

The statistical results of heavy metal concentrations in the street dust of QSD are shown in Table 1. The average concentrations of Cr, Mn, Ni, Zn, Fe, Cu, and Cd in the street dust were 70.17, 635.80, 22.17, 249.71, 5278.60, 51.15, and 0.86 mg/kg, respectively. It can be seen that the Ni concentration showed a wide range, and the difference between its minimum and maximum values was 58 times. According to the national limit guide for China [42], the concentrations of Cr, Ni, Cu, and Cd in the street dust did not exceed the limits, indicating that the concentration of heavy metals was low and there was no significant pollution. As there were no standard values for Mn, Fe, and Zn in the limit values, there was no basis by which to judge whether the two heavy metals exceeded the standard. The average concentrations of Cr, Mn, Fe, and Ni did not exceed the soil background value of Hubei Province [43], while the mean concentrations of Zn, Cu, and Cd were all higher than the soil background values, which were 2.99, 1.66 and 5 times that of the background value, respectively. In addition, due to industrial pollution, Zn is also derived from the abrasion of automobile tires and the leakage of lubricating oil, while Cu might also be related to traffic pollution emissions [44]. Cd is generally related to smelting activities or is derived from traffic pollution; thus, it may be affected by the activities of the Wuhan Iron and Steel (Group) Company (WISC). Therefore, transportation activities are the main reason for the high contents of Zn, Cu, and Cd in street dust from QSD.

The coefficient of variation (CV) of the heavy metals followed the decreasing order of Cd > Ni > Cr > Cu > Mn > Zn > Fe from small to large, with the coefficient of variation of Fe being 8.71%, indicating that the element was less affected by external factors and its spatial distribution was relatively uniform, with relatively similar pollution levels. The coefficients of variation of Cr, Mn, Zn, and Cu were in the range of 25~60%, while those of Ni and Cd were all higher than 100%. In particular, the CV value (319.09%) of Cd was the highest, indicating that these elements were subject to strong external interference and might be affected by human activities.

Table 2 summarizes the comparison of heavy metal concentrations in the street dust of QSD with that of other cities. Cr concentration in the street dust was lower than in samples taken in Beijing [45], Xi’an [46], Urumqi [47], and Panzhihua [48], which was comparable with those in Changsha [49], but higher than those in Suzhou [50] and Shenyang [51]. With the exception of Luanda [52] and Mexico [53], Cr concentration was at a lower level than in Dhaka [54], Kolkata [55], Ho Chi Minh City [56], and Bandar Abbas [57]. As for Mn, its concentration was relatively higher than those concentrations in other foreign cities. The Ni concentration was only higher than that found in Suzhou [49] and Lunda [50]. Zn was at a relatively low concentration, compared with other cities [46,48,50,52,56]. Cu and Cd concentrations were at a relatively low level, compared to those in other cities in China [47,50]. With the exception of Luanda [52] and Dhaka [54], Cu concentration was at a lower level than in other foreign cities. Cd concentration was only higher than that observed in Ho Chi Minh City [56] and Bandar Abbas [57]. The reason for this difference might be that it was greatly affected by different pollution sources.

### 3.2. Spatial Distribution of Heavy Metal Content

The spatial distribution map of heavy metal Cr, Mn, Ni, Zn, Fe, Cu, and Cd contents in the street dust of QSD in Wuhan City was drawn using the Surfer 10.0 program (Figure 2). It can be seen that the concentrations of Zn, Mn, and Fe show a trend of gradual increase from west to east. The WISC is located in the east of the study area. Fe is an essential element in the production of steel alloys. In the metallurgical industry, Mn is often used to make special types of steel, while ferromanganese alloy can also be used as a sulfur and oxygen remover in the production of steel. In addition, large trucks and lorries often drive on the roads around the WISC. The combustion of gasoline and diesel, as well as tire wear, will release particulates containing Zn [57]. Therefore, the concentrations of Zn, Mn, and Fe in street dust may be affected by both steel industry production and road transportation.

Cd concentrations in the southeast direction of the study area are extremely high, which is possibly from traffic emissions because Cd concentrations were greatly affected by automobiles, and the traffic volume in this section is mostly heavy trucks [58]. It could also be seen from the data that Mn and Cd are observed in high concentrations at the same point. This point is near a highway, and previous studies have also shown that the increase in Mn concentrations in road dust is related to the high road traffic flow [59].

There were four high-value points of Cu concentrations; two of them in the southeast direction might be closely related to the industrial production of the WISC. The other two high-value points are located in the northwest and southwest directions, where Linjiang Avenue and Huanle Avenue are located, respectively. The traffic flow is large, and the frequent human disturbances and motor vehicle brake wear emissions will lead to an increase in Cu concentrations in street dust [60]. Meanwhile, Zn, Fe, and Cu show multicenter points. These centers are located in the factory and road trunk roads, which together comprise a production–to–transport process, further indicating that the possible sources of these metals are related to industrial emissions from road transport.

The distributions of Ni and Cr are relatively similar, with high values appearing on Linjiang Avenue in the northwest. It is speculated that the high values may be related to the construction activities of newly built parks and the freight of heavy trucks, resulting in high concentrations of Ni and Cr.

### 3.3. Heavy Metal Pollution Evaluation

#### 3.3.1. Geo-Accumulation Indexes (*I*_geo_)

The *I*_geo_ values for heavy metals were calculated to further evaluate the pollution levels of heavy metals (Figure 3). The *I*_geo_ value of Cd in 47% of samples exceeded 1.0, indicating that the street dust was contaminated with Cd. The *I*_geo_ value of Zn exceeded 1.0 for 47% of the samples, indicating that Zn pollution was of a light to moderate degree. For Cu, only 10% of the samples had *I*_geo_ values above 1.0, indicating that the Cu contamination was slight. The average *I*_geo_ values of Cr, Mn, Fe, and Ni were all less than 0, indicating that Cr, Mn, Fe, and Ni were at a pollution-free level, with no significant enrichment.

#### 3.3.2. Potential Ecological Risk Assessment

Table 3 shows that the *E_r_^i^* values of heavy metals were in the order of Cd > Mn > Cu > Zn > Ni > Cr, and the *E_r_^i^* values of Cr, Mn, Ni, Zn, and Cu were all less than 30, indicating that these metals presented a slight ecological risk. The mean *E_r_^i^* value of Cd was the largest (149.52), indicating a strong ecological hazard. The *PERI* values of heavy metals were in the order of Cd > Mn > Cu > Zn > Ni > Cr, while the *PERI* value of Cr was less than 80, indicating that it was a minor ecological hazard. The *PERI* values of Ni and Zn were in the range of 80–160, indicating a medium ecological hazard. The *PERI* values of Mn, Cu, and Cd were all greater than 320 and indicated a strong ecological hazard. Mn, Cu, and Cd present a relatively strong ecological risk, which may be related to particulate matter that is emitted from motor vehicle fuel combustion and steel industry production. 

### 3.4. Principal Component Analysis

The results of the principal component analysis (PCA) (Figure 4) showed that the heavy metal pollution source was primarily maintained by two principal components (PCs) with a cumulative variance contribution rate of 69.75%. The contribution rate of PC 1 is 48.13%, and the factor loads of Cu, Cr, Mn, Zn, Ni, and Fe are all higher than 0.5. It is generally judged that these elements might be derived from a mixture of natural and anthropogenic sources. The wearing down of metal automobile parts might release Cu and Zn [61]. The combustion of car mixers or carburetors and production dust from smelting plants may also be a source of Cr in street dust [62]. Fe and Mn are indispensable elements in steel alloys and special steels. The WISC is the first ultra-large steel complex built after the founding of New China, based in the QSD in Wuhan City. The emission of tailpipe gas and dust during production, as well as the transportation of steel, may have a certain impact on soil particles in the surrounding environment, thereby affecting street dust as well. Therefore, PC1 might represent both industrial and transportation pollution.

The contribution of PC2 was 21.62%, with higher factor loads of 0.787, 0.539, and 0.492 for Ni, Cr, and Cd, respectively. Cr and Cd are generally considered to be related to transportation conditions, and their main impacts are derived from transportation exhaust [63]. Ni is closely related to metal smelting [64]; atmospheric dust will bring such heavy metals into the street dust. Therefore, PC2 might be attributed to traffic emissions and traffic carry-ons.

### 3.5. Health Risk Assessment

The non-carcinogenic health risk index (*HQ*), total non-carcinogenic risk index (*HI*), and carcinogenic risk index (*CR*) of heavy metals were calculated using the USEPA health risk assessment model. Table 4 shows these three data values. The results showed that the *HQ* and *HI* values of heavy metals were less than 1 in this study. This shows that these non-carcinogenic metals present no non-carcinogenic health risks to children and adults. However, compared with adults, children’s exposure to heavy metals in street dust had a higher non-carcinogenic risk. Some scholars believed that compared with adults, children’s higher intake and lower tolerance to pollutants will lead to such a situation [65]. Among these studied metals, Cr has the highest non-carcinogenic risk, followed by Cu and Zn. From the perspective of exposure routes for both adults and children, hand-to-mouth intake is the main exposure route of human contact with street dust, followed by skin contact exposure and respiratory inhalation. This finding has been confirmed in previous studies [29].

In this study, only Cr, Ni, and Cd exposure by inhalation were identified as carcinogenic. The carcinogenic risk results of these three heavy metals are shown in Table 4. The *CR* values of Cr, Ni, and Cd are all less than 1 × 10^−6^, indicating that there is no carcinogenic risk.

## 4. Conclusions

The pollution levels, spatial distribution, and potential risks of heavy metals in street dust were investigated in detail in the Qingshan District of Wuhan. The concentrations of Cr, Ni, Zn, Cu, and Cd all exceeded the national standard limit, and the enrichment of heavy metals was high. The *I*_geo_ values showed that Zn and Cu presented light and moderate pollution, while Cd showed moderate pollution. The potential ecological risk assessment showed that heavy metals posed medium to high potential ecological risks. The spatial distribution of heavy metals in street dust was consistent with the distribution of the surrounding traffic pollution sources and the pollution emission sources of the steel industry. The PCA results showed that heavy metal pollution was mainly affected by anthropogenic activities, and Cu, Cr, Zn, and Mn were mainly emitted from industrial processes. Ni, Cr and Cd were mainly from traffic pollution. Moreover, Cr, Ni, Zn, Cu, and Cd did not pose any non-carcinogenic health risk to adults and children; their risk levels to adults and children are shown in descending order: Cr > Cu > Ni > Cd > Zn. The carcinogenic risk of Cr, Ni, and Cd was less than 1 × 10^−6^, indicating that there was no carcinogenic risk. Based on the spatial distribution and PCA results, it is necessary to manage industrial and traffic emissions more strictly in order to reduce the pollution risk caused by heavy metals in the future. Overall, the present study results might be very helpful to better develop future management risk strategies for urban environments, especially for metropolitan areas in China.

## Figures and Tables

**Figure 1 ijerph-19-10970-f001:**
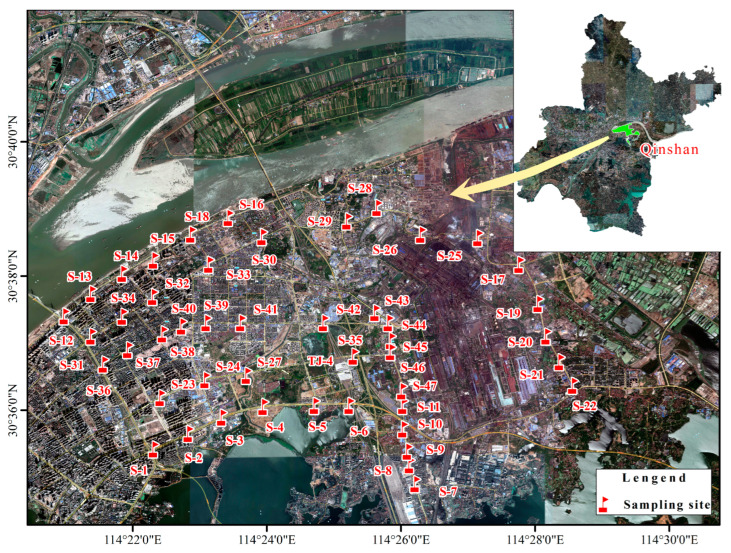
Map of sampling sites in the Qingshan District of Wuhan City.

**Figure 2 ijerph-19-10970-f002:**
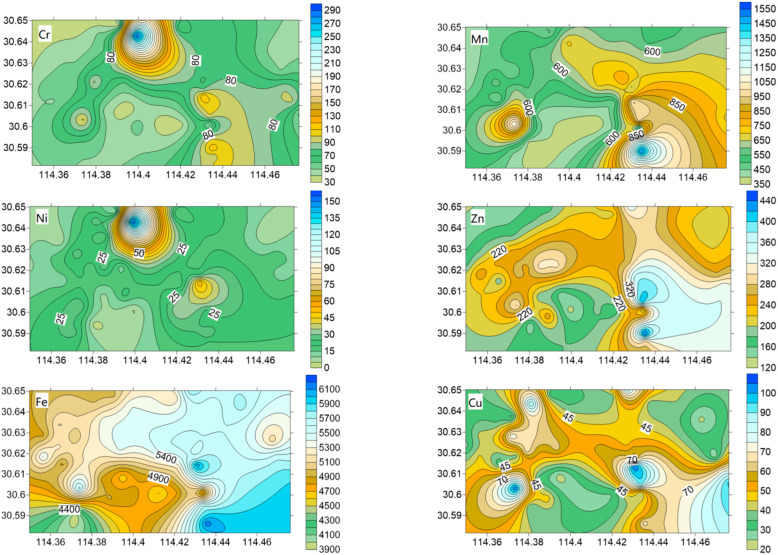
Spatial distribution of heavy metal contents in the street dust of Qingshan District in Wuhan City.

**Figure 3 ijerph-19-10970-f003:**
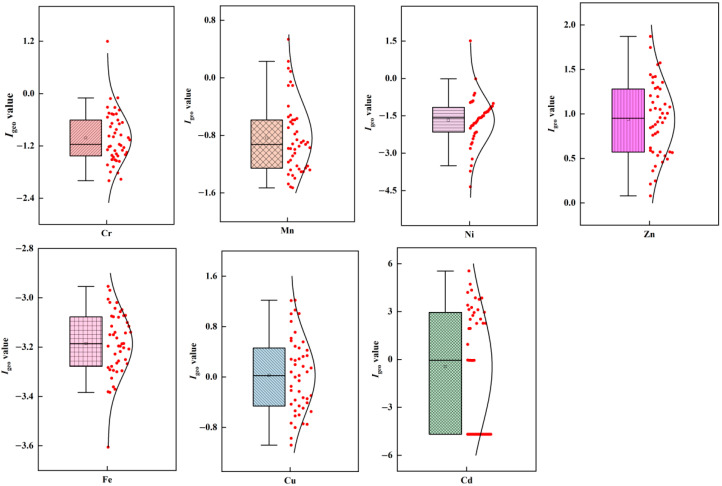
The contamination of heavy metals indicated by the geoaccumulation index (*I*_geo_) of the dust.

**Figure 4 ijerph-19-10970-f004:**
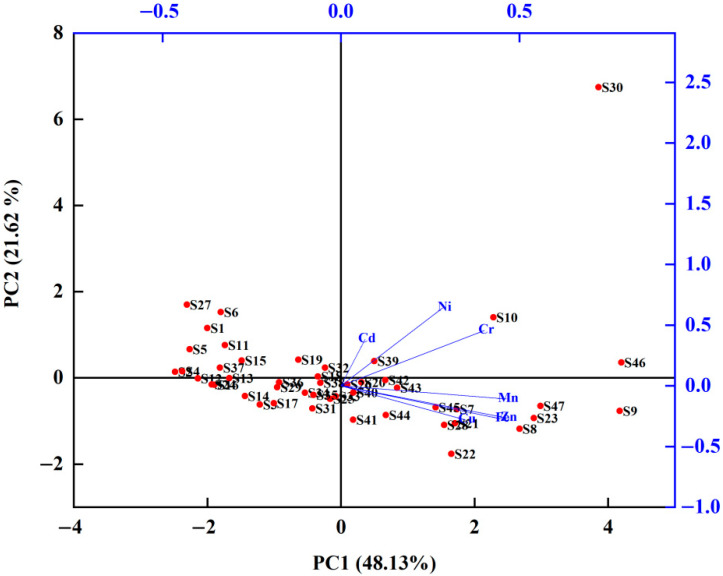
Factor loadings of principal component analysis of heavy metals in street dust.

**Table 1 ijerph-19-10970-t001:** Statistical description of heavy metal concentrations (mg/kg) in the street dust of the Qingshan District.

Metal	Minimum	Maximum	Mean Value ± SD	CV (%)	Background Value ^a^	National Guide Value ^b^
Cr	32.25	295.94	70.17 ± 40.74	58.05	86	78
Mn	370	1545.90	635.80 ± 256.23	40.30	712	—
Ni	2.74	158.74	22.17 ± 22.46	101.32	37.3	2000
Zn	132.49	458.76	249.71 ± 73.48	29.42	83.6	—
Fe	3931.37	6176.01	5278.60 ± 265.23	8.71	39,100	—
Cu	21.76	107.20	51.15 ± 22.37	43.74	30.7	36,000
Cd	0.001	12.01	0.86 ± 2.74	319.09	0.172	172

SD: standard deviation. CV: coefficient of variation. ^a^ The background values of soil in Hubei Province, China [43]; ^b^ the guide values based on the soil environmental quality risk control standards for the soil contamination of development land (GB36600-2018) [42].

**Table 2 ijerph-19-10970-t002:** Comparative study of heavy metal concentrations in street dust between Wuhan and other cities.

City	Heavy Metal Content in Street Dust (mg/kg)							Reference
	Cr	Mn	Ni	Zn	Fe	Cu	Cd	
Wuhan, China	70.17	635.80	22.17	249.71	5278.6	51.15	0.86	This study
Beijing, China	92.10	553.73	32.47	280.65	—	83.12	0.59	[45]
Xi’an, China	167.28	687	—	421.46	—	94.98	—	[46]
Suzhou, China	25.70	—	16.40	376.90	—	104.80	2.45	[50]
Urumqi, China	186.00	—	289.70	227.00	—	179.00	1.97	[47]
Changsha, China	71.6	—		171.00	21,500	43.90	7.48	[49]
Shenyang, China	40.17	—	35.11	140.24	—	41.19	0.37	[51]
Panzhihua, China	228	—	62.5	373	—	68.6	0.96	[48]
Kolkata, India	114.00	543.00	51.00	249.00	114.00	466.90	—	[55]
Luanda, Angola	26.00	258.00	10.00	317.00	11,572.00	42.00	1.10	[52]
Dhaka, Bangladesh	144.34	261.53	37.01	239.16	—	49.68	11.64	[54]
Ho Chi Minh City, Vietnam	102.4	393.9	36.2	466.4	—	153.7	0.5	[56]
Bandar Abbas, Iran	73.51	458.75	65.97	292.92	—	149.75	0.42	[38]
Mexico City, Mexico	51.4	235.2	36.3	280.7	5722.2	99.7	—	[53]

**Table 3 ijerph-19-10970-t003:** Potential ecological risk index (*PE**RI*) values of heavy metals in street dust.

Element	$E_{r}^{i}\;(\mathrm{Grading})$			*PERI*	Risk Level
	Mix Value	Min Value	Ave Value		
Cr	0.75	6.88	1.63	76.70	Slight ecological risk
Mn	5.20	21.71	8.93	419.70	Strong ecological hazard
Ni	0.37	21.28	2.97	139.67	Medium ecological hazard
Zn	1.58	5.49	2.99	140.39	Medium ecological hazard
Cu	3.55	17.45	8.32	391.50	Strong ecological hazard
Cd	−532.52	2095.12	149.52	7027.28	Strong ecological hazard

**Table 4 ijerph-19-10970-t004:** Non-carcinogenic risk and the carcinogenic risk index of heavy metals in street dust.

Element	*HQ* _ing_		*HQ* _inh_		*HQ* _dermal_		*HI*		*CR*
	Adult	Child	Adult	Child	Adult	Child	Adult	Child	
Cr	3.75 × 10^−2^	2.68 × 10^−1^	3.82 × 10^−4^	8.13 × 10^−4^	2.23 × 10^−3^	1.20 × 10^−2^	4.01 × 10^−2^	2.81 × 10^−1^	2.41 × 10^−7^
Ni	1.78 × 10^−3^	1.27 × 10^−2^	1.67 × 10^−7^	3.56 × 10^−7^	7.84 × 10^−6^	4.23 × 10^−5^	1.79 × 10^−3^	1.27 × 10^−2^	1.52 × 10^−9^
Cd	1.37 × 10^−3^	9.82 × 10^−3^	2.33 × 10^−6^	4.97 × 10^−6^	6.55 × 10^−5^	3.53 × 10^−4^	1.44 × 10^−3^	1.02 × 10^−2^	4.41 × 10^−10^
Cu	2.05 × 10^−3^	1.47 × 10^−2^	1.98 × 10^−7^	4.21 × 10^−7^	8.14 × 10^−6^	4.39 × 10^−5^	2.06 × 10^−3^	1.47 × 10^−2^	
Zn	1.33 × 10^−3^	9.54 × 10^−3^	1.29 × 10^−7^	2.76 × 10^−7^	7.95 × 10^−6^	4.29 × 10^−5^	1.34 × 10^−3^	9.58 × 10^−3^	
Total	4.41 × 10^−2^	3.15 × 10^−1^	3.84 × 10^−4^	8.19 × 10^−4^	2.32 × 10^−3^	1.25 × 10^−2^	4.68 × 10^−2^	3.28 × 10^−1^	2.43 × 10^−7^

## Data Availability

The data used to support the findings of this study are available from the corresponding author.

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
