# Peer review of "Pollution Characteristics and Human Health Risk Assessment of Heavy Metals in Street Dust from a Typical Industrial Zone in Wuhan City, Central China"

_ijerph, 2022, doi:10.3390/ijerph191710970_

Round 1

Reviewer 1 Report

Title: Status, sources, and human health risk assessment of heavy metals in street dust from a typical industrial zone of Wuhan City

Journal: International Journal of Environmental Research and Public Health

Comments: In this study, contents of six heavy metals including Cr, Mn, Ni, Zn, Fe, Cu, and Cd in the street dust collected from Qingshan District, Wuhan city, China were quantified. The spatial heterogeneity and associated fountainhead were elucidated based on the spatial distribution map establishment and PCA. In addition, ecological risk assessment indicated the medium to high risk level posed by the heavy metals in QSD. The carcinogenic health risk induced by the heavy metals to adults and children was insignificant. The experiment and analysis were sufficient, and the aims were explicit. However, there were some comments for the descriptions and the discussion. The main comments were as below:

(1) Abstract section, the concrete values for the concentrations of heavy metals should show in this section, and the comparison with other zones or countries was meaningful. Please add it.

(2) Line 20-21, the description here was blurry. Please emphasize it.

(3) Line 25, the values range of RI should be added.

(4) Line 233-235, as described previously, the concentrations of Cr in this study were lower than Beijing, Xian, and Urumqi, but the authors stated it was at a higher level than other cities worldwide. Its confused. Moreover, the other cities in here need a reference.

(5) Line 238-240, the occurrence level of Zn, Cu, and Ca showed similar rank compared with other investigations. Its better to combine the descriptions to avoid redundancy. The references associated with Zn and Cu were missing here.

(6) Section 3.2. the spatial distribution map was interesting. However, the analysis about it was superficial. There were many information, such as, the vertex of Mn and Cd were observed at the same point, and if there was a factory here? For Zn, Fe, and Cu, there were multicenter found, and the underlying reasons might be interesting. Please polish this section carefully.

(7) Figure 3, box plot equipped with percentiles were recommended here. If that, the range of different samples was visual.

(8) Line 272, there were double traffic flow.

(9) Line 305, particulate matter?

(10) Section 3.4, the PCA plot with scatter points and arrow heads were strongly recommended here. If that, the readers can visually discriminate the variation and cluster of different heavy metals. Moreover, the descriptions about why the first principal components included Cu, Cr, Zn, Fe and Mn was unclear. The authors should combine the context to further explain it, because its a highlight for this study.

(11) Line 351, Table 9?

(12) Conclusion Section didnt aim to conclude the results, and prone to state the highlight, perspectives, or insights of this study based on the results and discussion. Hence, its better to reconstruct this section.

Author Response

Answer to Reviewer #1:

General comments: In this study, contents of six heavy metals including Cr, Mn, Ni, Zn, Fe, Cu, and Cd in the street dust collected from Qingshan District, Wuhan city, China were quantified. The spatial heterogeneity and associated fountainhead were elucidated based on the spatial distribution map establishment and PCA. In addition, ecological risk assessment indicated the medium to high risk level posed by the heavy metals in QSD. The carcinogenic health risk induced by the heavy metals to adults and children was insignificant. The experiment and analysis were sufficient, and the aims were explicit. However, there were some comments for the descriptions and the discussion. The main comments were as below:

Q1: Abstract section, the concrete values for the concentrations of heavy metals should show in this section, and the comparison with other zones or countries was meaningful. Please add it.

Reply: Considering the reviewer’s suggestion, we have thoughtful revised Abstract section, please seen the revised manuscript. In addition, we have revised the abstract of this manuscript, where the primary purpose, the main experimental methods, the principal results, the major conclusions and the great significance of this research have been elaborated in detail. For easy check/editing purposes, the extensive modification sections were all marked as red font in the revised manuscript.

Q2: Line 20-21, the description here was blurry. Please emphasize it.

Reply: Considering the reviewer’s suggestion, the sentence “Results showed that except for Cr, Mn, Fe, and Ni, the mean concentrations of Zn, Cu, and Pd in dust were higher than the soil background value in Hubei Province. The enrichment of heavy metals varied in different streets, which is closely related to urban transportation and industrial production.” have revised to “Results indicated that the mean concentrations of Zn, Cu, and Pd in street dust were higher than their corresponding soil background values in Hubei Province. heavy metal enrichment is closely related to urban transportation and industrial production”

Q3: Line 25, the values range of RI should be added.

Reply: In accordance with your advice, the values range of RI have been added in Abstract section. Please seen the sentence below:

The PERI values of heavy metals in street dusts ranged between 76.70 and 7027.28, which has posed a medium to high potential ecological risk.

Q4: Line 233-235, as described previously, the concentrations of Cr in this study were lower than Beijing, Xi’an, and Urumqi, but the authors stated it was at a higher level than other cities worldwide. Its confused. Moreover, the other cities in here need a reference.

Reply: In response to this problem, the sentence “With the exception of Dhaka (Safiur Rahman et al., 2019) and Kolkata (Kolakkandi et al., 2020), Cr concentration in this study was at a higher level than in the other cities worldwide” have revised to “With the exception of Luanda (Ferreira-Baptista and De Miguel, 2005) and Mexico (Aguilera et al., 2021), Cr concentration in this study was at a lower level than Dhaka (Safiur Rahman et al., 2019), Kolkata (Kolakkandi et al., 2020), Ho Chi Minh (Dat et al., 2021) and Bandar Abbas (Heidari et al., 2021).”

Q5: Line 238-240, the occurrence level of Zn, Cu, and Ca showed similar rank compared with other investigations. Its better to combine the descriptions to avoid redundancy. The references associated with Zn and Cu were missing here.

Reply: Considering the reviewer’s suggestion, the sentence “Zn was at a relatively low concentration compared with other cities. Cu and Cd concentration were at a relatively low level compared to those in other cities in China, but compared to foreign cities, with the exception of Luanda (Ferreira-Baptista and De Miguel, 2005), Dhaka (Safiur Rahman et al., 2019), Cu concentration in this study at a low level than other foreign cities in Table 2. the concentration of Cd was only higher than that observed in Ho Chi Minh (Dat et al., 2021), Bandar Abbas (Heidari et al., 2021).”

Q6: Section 3.2. the spatial distribution map was interesting. However, the analysis about it was superficial. There were many information, such as, the vertex of Mn and Cd were observed at the same point, and if there was a factory here? For Zn, Fe, and Cu, there were multicenter found, and the underlying reasons might be interesting. Please polish this section carefully.

Reply: Considering the reviewer’s suggestion, the sentence “It can also be seen that Mn and Cd are observed in high concentration at the same point. This point is near a highway, and previous studies have also shown that the in-crease of manganese concentration in road dust is related to the high road traffic flow (Al-Taani et al. 2019 and Anna Bourliva).” “Meanwhile, Zn, Fe and Cu show multicenter points in the figure. These centers are lo-cated in the factory and road trunk roads, which together appear to be a production to transport process, further indicating that the possible source of these metals is related to industrial emissions from road transport..”

Q7: Figure 3, box plot equipped with percentiles were recommended here. If that, the range of different samples was visual.

Reply: In line with the reviewer’s comment, we have offered the box plot of geoaccumulation index (Igeo) values of heavy metals with percentiles shown in Page 9, line 295.

Q8: Line 272, there were double “traffic flow”.

Reply: Considering the reviewer’s suggestion, “The traffic flow and traffic flow are large” in line 276 have been revised to “The traffic flow are large”.

Q9: Line 305, “particulate matter”?

Reply: In accordance with your advice, “particulate matt ” have been revised to “particulate matter” in line 310.

Q10: Section 3.4, the PCA plot with scatter points and arrow heads were strongly recommended here. If that, the readers can visually discriminate the variation and cluster of different heavy metals. Moreover, the descriptions about why the first principal components included Cu, Cr, Zn, Fe and Mn was unclear. The authors should combine the context to further explain it, because its a highlight for this study.

Reply: In accordance with your advice, the first principal components accounting for 26.66 % of the total variance showed positive loadings on Cu, Cr, Mn, Zn, Ni and Fe, due to their loadings more than 0.5. It is generally judged that these elements might be mainly derived from a mixture of natural and anthropogenic sources.

Q11: Line 351, Table 9?

Reply: In line with the reviewer’s comment, Table show in Page 4, line 355 should be “Table 4”.

Q12: Conclusion Section didn’t aim to conclude the results, and prone to state the highlight, perspectives, or insights of this study based on the results and discussion. Hence, its better to reconstruct this section.

Reply: In accordance with your advice, we have thoughtful revised Conclusion section, please seen the revised manuscript. Meanwhile, we point out the future implications of this study, which might be very helpful to better develop management risk strategies for metropolis in China. In addition, this paper been nicely written in well-structured manner and further supported with study data and previous literature data. We have improved the discussion part by adding more supporting data from the literature. Moreover, we have tried to offer a higher quality of paper for global readers in the field. We have also revised the problematic parts of this manuscript, where the primary purpose, the principal results, major conclusions and great significance of this research have been elaborated in detail. In last, we improve manuscript draft in English language. 

Reviewer 2 Report

-       Page 3: Identification in the legend of Figure 1 of the symbology used, for example, LJ-1, HP-1 etc.

-       Page 4, line 135: “Where Cn is the content of heavy metal in street dust and Bn is the geochemical background value of the heavy metal.“ should be “Where Cn is the concentration of heavy metal in street dust (mg/kg) and Bn is the geochemical background concentration of the heavy metal (mg/kg). “

-       Page 4, line 152; “Where, Ci is the calculated contamination factor; Cr is the measured content of each heavy metal and Cnis the geological background value of the heavy metal;” should be “Where, Ci is the calculated contamination factor; Cr is the measured concentration of each heavy metal (mg/kg) and Cnis the geological background concentration of the heavy metal (mg/kg);”

-       Page 9, line 313: “The contribution rate of principal component 1is 48.34%, and…” should be “The contribution rate of principal component 1 is 48.34%, and…”

Author Response

Answer to Reviewer #2:

General comments: Comments and Suggestions for Authors

Q1: Page 3, Identification in the legend of Figure 1 of the symbology used, for example, LJ-1, HP-1 etc.

Reply: In accordance with your advice, we have added the meanings of symbology used in Figure 1.

Q2: Page 4, line 135: “Where Cn is the content of heavy metal in street dust and Bn is the geochemical background value of the heavy metal.“ should be “Where Cn is the concentration of heavy metal in street dust (mg/kg) and Bn is the geochemical background concentration of the heavy metal (mg/kg)”.

Reply: Considering the reviewer’s suggestion, the sentence “Cn is the content of heavy metal in street dust and Bn is the geochemical background value of the heavy metal” have been thoughtfully revised to “Cn (mg/kg) is the concentration of each metal in street dust and Bn is the geochemical background concentration (mg/kg) of each metal”.

Q3: Page 4, line 152; “Where, Cif  is the calculated contamination factor; Cri  is the measured content of each heavy metal and Cni is the geological background value of the heavy metal;” should be “Where, Cif  is the calculated contamination factor; Cri  is the measured concentration of each heavy metal (mg/kg) and Cni is the geological background concentration of the heavy metal (mg/kg);”

Reply: Considering the reviewer’s suggestion, the sentence “Cif is the calculated contamination factor; Cri  is the measured content of each heavy metal and Cni is the geological background value of the heavy metal” have been thoughtfully revised to “Cri (mg/kg) is the measured concentration of each metal and Cni (mg/kg) is the geological background value of each metal”.

Q4: Page 9, line 313: “The contribution rate of principal component 1is 48.34%, and…” should be “The contribution rate of principal component 1 is 48.34%, and…”

Reply: Following the reviewer’s comment, the sentence “The contribution rate of principal component 1is 48.34%” have been thoughtfully revised to “The contribution rate of principal component 1 is 48.34%, and…”

Round 2

Reviewer 1 Report

The authors have answered all my questions, and there are no more comments. Its be recommended accepted at present form.